# Evaluating the Effectiveness of Calcium Silicate in Enhancing Soybean Growth and Yield

**DOI:** 10.3390/plants12112190

**Published:** 2023-05-31

**Authors:** John Quarshie Attipoe, Waleed Khan, Rupesh Tayade, Senabulya Steven, Mohammad Shafiqul Islam, Liny Lay, Amit Ghimire, Hogyun Kim, Muong Sereyvichea, Then Propey, Yam Bahadur Rana, Yoonha Kim

**Affiliations:** 1Department of Food Security and Agricultural Development, Kyungpook National University, Daegu 41566, Republic of Korea; jerryjohn2487@gmail.com (J.Q.A.); stevensenabulya@gmail.com (S.S.); sereyvicheamuong@gmail.com (M.S.); tpropey@gmail.com (T.P.); yamanrana2015@gmail.com (Y.B.R.); 2Laboratory of Crop Production, Department of Applied Biosciences, Kyungpook National University, Daegu 41566, Republic of Korea; waleedkhan.my@gmail.com (W.K.); rupesh.tayade@gmail.com (R.T.); shafik.hort@gmail.com (M.S.I.); layliny22@gmail.com (L.L.); ghimireamit2009@gmail.com (A.G.); rlfjrl1000@naver.com (H.K.); 3Upland Field Machinery Research Center, Kyungpook National University, Daegu 41566, Republic of Korea

**Keywords:** silicon fertilizer, soybean, vegetative indices, root traits, yield, GIS

## Abstract

The application of silicon (Si) fertilizer positively impacts crop health, yield, and seed quality worldwide. Si is a “quasi-essential” element that is crucial for plant nutrition and stress response but is less associated with growth. This study aimed to investigate the effect of Si on the yield of cultivated soybean (*Glycine max* L). Two locations, Gyeongsan and Gunwi, in the Republic of Korea were selected, and a land suitability analysis was performed using QGIS version 3.28.1. The experiments at both locations consisted of three treatments: the control, Si fertilizer application at 2.3 kg per plot (9 m × 9 m) (T1), and Si fertilizer application at 4.6 kg per plot (9 m × 9 m) (T2). The agronomic, root, and yield traits, as well as vegetative indices, were analyzed to evaluate the overall impact of Si. The results demonstrated that Si had consistently significant effects on most root and shoot parameters in the two experimental fields, which led to significantly increased crop yield when compared with the control, with T2 (22.8% and 25.6%, representing an output of 2.19 and 2.24 t ha^−1^ at Gyeongsan and Gunwi, respectively) showing a higher yield than T1 (11% and 14.2%, representing 1.98 and 2.04 t ha^−1^ at Gyeongsan and Gunwi, respectively). These results demonstrate the positive impact of exogenous Si application on the overall growth, morphological and physiological traits, and yield output of soybeans. However, the application of the optimal concentration of Si according to the crop requirement, soil status, and environmental conditions requires further studies.

## 1. Introduction

Soybean (*Glycine max L*.) is widely cultivated and is one of the most significant seed legumes in the world due to its high protein and oil contents, which make it an excellent nutritional source for both animals and humans [1,2]. Soybean generates approximately 23% and 25% of the global protein and oil supply, respectively [3]. Numerous studies focusing on improving soybean production are currently targeting its morphological, physiological, and genetic characteristics [2,4,5]. As a result, studies on the exogenous application of various nutrient fertilizers and their impact on general crop development and yield have been extensively prioritized, with Si currently being considered as a potential growth promoter. Silicon (Si) is the second major element after oxygen (O_2_) that occurs within the Earth’s crust, with an abundance of approximately 28% [6]. However, its role in growth and development or its effects on metabolism and physiological functions in plants have not been fully understood; thus, it is not considered an essential nutrient for crop production in the agricultural sector [7].

Si predominantly occurs in the Earth’s crust as Si dioxide (SiO_2_), and it is absorbed by plants in the soil solution as soluble mono-silicic acid (H_4_SiO_4_) [8,9]. In the rhizosphere, Si constitutes approximately 60% of the Earth’s crust [1,10]. Although Si has not been recognized as a necessary nutrient for plant growth, numerous studies have shown its positive impact on all the important growth parameters of a variety of crops, including both monocots and dicots, resulting in higher yields [11,12]. Such results have been demonstrated in a variety of crops, such as sugarcane (*Saccharum officinarum*) [13], rice (*Oryza sativa*) [14], maize (*Zea mays* L.) [15,16,17,18,19], and many others from the last decade with a focus on different morphological, physiological, and biochemical traits. Recent studies have shown the potential of Si in protecting crop plants against the adverse impacts of abiotic stressors. The uptake of Si by crop plants triggers the activation of various important genes that help alleviate the effects of stress and improve the regulation of plant growth and development [12]. In crop plants, a combination of specific and nonspecific transporters mediates the transport of Si into the aerial organs. *Lsi2*, *Lsi1*, and *Lsi6* expression primarily facilitates the uptake of Si in both the roots and aerial tissues of several crop species, including maize, cucumber (*Cucumus sativus* L.), rice, and barley (*Hordeum vulgare*) [20,21,22]. These genes are known to regulate the uptake of Si, thereby playing a pivotal role in enhancing the ability of plants to cope with abiotic stressors [23,24,25], of these transporters, aquaporin-based *Lsi1* and *Lsi6* are extensively distributed in the root and shoot tissues, whereas *Lsi2*, an anion transporter, is mainly present in the endodermal root membranes [26]. Although crop plants can thrive without Si, it was observed that plants, such as rice and horsetail (*Equisetum arvense*), may become more susceptible to fungal infections in the absence of Si [27]. Si has been found to exert multiple functions in alleviating abiotic stress conditions [28,29,30,31,32]. Si has been demonstrated to facilitate a variety of methods for sequestering metal ions, including coprecipitation, the modulation of soil pH, metal speciation, and compartmentalization [33]. Recent developments in Si-based fertilizers have demonstrated their potential in promoting the growth and development of crop plants by enhancing photosynthesis and regulating electrolytic leakage under stressful conditions [34]. Si has been shown to enhance photosynthesis and improve water and nutrient uptake in mango trees under the conditions of abiotic stress [35]. Furthermore, Si is present in variable quantities in nearly all plant species, exerting a variety of physiological effects [36]. In addition to its critical function in improving stress tolerance to salt and drought, Si uptake has been shown to enhance the mechanical support of shoots and leaf blades in plants [30,34]. Si has been widely studied for its effectiveness in mitigating both abiotic and biotic stresses across various plant species with different mechanisms [37,38]; Si plays a crucial role in ameliorating metal toxicity in numerous crop plants, particularly for metal ions, such as aluminum (Al) and manganese (Mn) [34,39,40,41]. Another study demonstrated that exogenous Si application aids in upregulating water uptake through aquaporins and root hydraulic conductance [42]. Moreover, Si supplementation in Talh trees (*Acacia gerrardii* Benth) renders them more tolerant to salinity, possibly through overproduction of glycine betaine and proline, which conserve water in tissues and positively regulate metabolic activity [43]. However, compared with other cereal crops, the effects of Si on legumes have received less attention [6,44]. Recent findings have shown that some legumes have the capacity to store significant quantities of Si in their leaf tissues [6,45]. Studies on the effects of Si in important leguminous crops have demonstrated significant positive results on key parameters, from shoot to root, and overall yield improvement. A recent study showed that Si positively impacted net photosynthesis and root and shoot morphologies as well as enhanced abiotic and biotic stress responses [46], thereby being dubbed as a “quasi-essential” nutrient [37,47].

Both dicotyledonous and monocotyledonous plants harbor Si transport genes. A previous study reported that influx Si transporter genes, including *GmNIP2-1* and *GmNIP2-2*, regulate the accumulation of Si in soybean and that Si is a beneficial element in soybean crops [45]. Moreover, a significant relationship between nitrogen (N) and Si in crops for yield response has been reported [48]. For example, the application of Si and N to soybean and common bean *(Phaseolus vulgaris* L.) enhanced plant development [49], whereas increased N_2_ fixation in symbiotic cowpea *(Vigna unguiculata)* led to higher nodule formation [50,51]. Abscisic acid production in the roots and lateral root development following Si application increased root growth in legumes and boosted their nodule size and quantity [48,52]. Further studies are needed to determine the precise application technique and optimum Si concentration. There is relatively limited information on Si buildup or its effects on soybean growth and production. Si can be applied exogenously to the soil or the foliar, and each application method has been shown to exhibit merits and demerits [53,54]. Foliar applications of Si have been demonstrated to enhance plant growth, yield, endogenous silicon content, and responses to biotic and abiotic stresses in various crops, such as rice [55], finger millet *(Eleusine coracana*) [56], maize [57], grape (*Vitis vinifera*) [58], coffee (*Coffea arabica*) [59], cucumber [60], and soybean [3]. A recent study that followed the effects of Si fertilizer on soybeans for 2 consecutive years (2018 and 2019) found significant positive effects on all important shoot and root parameters as well as an increase in grain production in the two seasons by 20% and 10%, respectively [2]. The present study was conducted in a controlled greenhouse environment.

The application of Si fertilizer significantly improves the growth and development of crops and increases their yield output. The objectives of this study were to conduct field-based experiments to evaluate the effects of a commercial silt fertilizer (super silicic acid) on key soybean root and shoot characteristics and compare its performance on the overall growth and output, i.e., yield, across different environments.

## 2. Results

An analysis of variance (ANOVA) showed significant differences (*p* < 0.001) in the 18 measured traits, except for the photochemical reflectance index (PRI), which only demonstrated a significant difference for different days of data collection (DC). The normalized difference between vegetative index (NDVI) and chlorophyll content (Chl) showed statistically significant differences (*p* < 0.0001) among different treatments, DC, and environments/locations (Table 1). Similarly, shoot traits, including plant height (PH) and the number of branches (NB), showed statistically significant differences (*p* < 0.0001) among different treatments, DC, and environments/locations (Table 1). The study revealed significant variations in important plant traits, such as those related to root architecture and yield. In particular, measures such as total root length (TRL), root surface area (SA), average root diameter (AD), root volume (RV), number of tips (NT), and number of forks (NF) exhibited statistically significant differences (*p* < 0.0001) among different treatments. Similarly, yield attributes, such as hundred seed weight (HSW), weight per grain (WPG), total grain weight (TGW), total grain number (TGN), and pod number (PN), were significantly distinct among different treatments. Notably, a few of these traits were also significantly different among different environmental conditions and treatment-environment interactions (Table 1).

### 2.1. Effects of Si Treatment on Soybean Plant Attributes

A posthoc Duncan test was performed for all traits that showed significant differences after treatment. As a result, T2 was found to positively influence the root, shoot, and yield traits, irrespective of the location (Figure 1, Figure 2, Figure 3 and Figure 4). The results for VIs and chlorophyll content on the second DC 79 days after planting (DAP) were considerably different. Figure 1a shows that at Gyeongsan, NDVI showed an increase of 12.3% between the control and T1 and 7.2% between the control and T2. At Gunwi, NDVI showed an increase of 5.77% and 9.29% for T1 and T2, respectively (Figure 1d). Unlike NDVI, PRI showed inconsistent values, with an increase at Gyeongsan, whereas this was relatively insignificant to the control at Gunwi (Figure 1b,e). At Gyeongsan (Figure 1c), the chlorophyll content showed a considerable increase of 20.12% and 29.89% for T1 and T2, respectively, compared with the control, whereas at Gunwi, it showed a significant increase of 23.32% and 33.16% for T1 and T2, respectively (Figure 1f).

Changes in the shoot traits were more evident at 93 DAP in DC for NB. At Gyeongsan, significant increases of 15.5% and 16.5% were observed for T1 and T2, respectively, relative to the control (Figure 2a), whereas at Gunwi, an increase of 29.6% for T1 and 29.09% for T2 was observed (Figure 2d). Regarding PH, an increase of 3.4% and 7.1% was observed for T1 and T2, respectively, relative to the control at Gyeongsan, whereas it increased by 5.9% for T1 and 6.5% for T2 at Gunwi (Figure 2b,e). Similarly, SW showed an increase of 2.24% and 12.6% for T1 and T2, respectively, relative to the control at Gyeongsan (Figure 2c), whereas it increased by 13% and 20% for T1 and T2, respectively, relative to the control at Gunwi (Figure 2f).

Regarding the key root traits, our results revealed that TRL increased by 28% and 41% for T1 and T2, respectively, relative to the control at Gyeongsan (Figure 3a), whereas it increased by 43% and 90% for T1 and T2, respectively, relative to the control at Gunwi (Figure 3a). Similarly, SA increased by 33% and 45% for T1 and T2, respectively, relative to the control at Gyeongsan and by 29% and 53% for T1 and T2, respectively, at Gunwi (Figure 3b). Moreover, NT increased by 50% and 78% for T1 and T2, respectively, at Gyeongsan, whereas it increased by 47% and 191% for T1 and T2, respectively, at Gunwi (Figure 3c). Since AD is negatively correlated with TRL, a reduction of −25% and −30% in its values was observed at Gyeongsan for T1 and T2, respectively, and a reduction of −15% and −29% was observed for T1 and T2 at Gunwi, respectively (Figure 3d). RV was also found to increase with Si treatment, with an increase of 20% and 36% for T1 and T2, respectively, at Gyeongsan and 19% and 42% for T1 and T2, respectively, at Gunwi (Figure 3e). NF increased by 35% and 60% for T1 and T2, respectively, at Gyeongsan, whereas it increased by 39% and 185% for T1 and T2, respectively, at Gunwi (Figure 3f).

Si treatment also exerted considerable effects on the yield traits. For PN, a notable increase of 30% and 46% was observed for T1 and T2, respectively, at Gyeongsan, whereas its values increased by 9.54% and 53% for T1 and T2, respectively, at Gunwi (Figure 4a). TGN increased by 28% and 62% for T1 and T2, respectively, at Gyeongsan and by 3% and 41% for T1 and T2, respectively, at Gunwi (Figure 4b). Similarly, TGW increased by 9.9% and 50% for T1 and T2, respectively, at Gyeongsan and by 12% and 25% for T1 and T2, respectively, at Gunwi (Figure 4c). Interestingly, WPG increased by 8% for T1 but decreased by 11% for T2 at Gyeongsan, whereas it decreased by 14% and 7.6% for T1 and T2, respectively, at Gunwi (Figure 4d).

HSW is negatively correlated with PN and TGN, and its values decreased with all treatments at both locations, with a 15% and 11% decrease at Gyeongsan and Gunwi, respectively (Figure 4e). A significantly different yield output was detected for T1, i.e., an 11% increase, representing 1.98 t ha^−1^, whereas an increase of 22.8%, representing 2.19 t ha^−1^, was detected in T2, relative to the control at Gyeongsan (Figure 4f). Similarly, a notable increase of 14.2% for T1, representing an output of 2.04 tons yield per hectare at Gunwi, and a significant 25.6% increase for T2, representing an output of 2.24 tons per hectare relative to the control, was observed (Figure 4f).

### 2.2. Correlations between Yield and Root and Shoot Traits

In order to determine the possible relationships between the yield, shoot, and root traits, a Pearson’s correlation test was performed separately at both locations. For root traits at both locations, TRL showed a positive correlation with SA, whereas the opposite was observed for root AD (Figure 5). At Gyeongsan, TRL exhibited varied relationships with important yield traits, i.e., it exhibited a negative correlation with HSW and WPG and a positive correlation with TGN and PN (Figure 5a). Among the yield traits, HSW showed a strong positive correlation with WPG and a negative correlation with TGN and PN. The chlorophyll content was negatively correlated with HSW and WPG but positively correlated with PN and TGN. PH showed a positive correlation with PN and TGN and a weak negative correlation with WPG and HSW. In addition, a positive correlation was observed between TRL and PH (Figure 5a).

At Gunwi, TRL was negatively correlated with HSW and WPG but strongly positively correlated with PN and TGN (Figure 5b). HSW showed a strong correlation with WPG but a negative correlation with PN and TGN. PH showed a positive correlation with PN and TGN but a negligible positive correlation with WPG and HSW. TRL was found to be positively correlated with PH (Figure 5b).

Pearson correlation coefficients between yield traits and combined total yield were evaluated at both experimental locations (Figure 6). The result showed a positive impact of PN and TGN on total yield. On the other hand, the relationship between HSW and yield showed a weak negative correlation (Figure 6). Overall, the correlation analysis revealed that TGN, PN, and HSW are important yield traits, with PN and TGN being the most crucial traits contributing to total crop yield.

## 3. Discussion

This study aimed to evaluate the efficacy of different Si fertilizer doses on soybean yield in a field experiment by comprehensively assessing all the key root, shoot, and yield traits. Our findings suggest that Si application positively impacts almost all important traits, ultimately leading to a significant increase in soybean yield. Our results revealed considerable improvements in shoot morphological and physiological traits, including Chl, NB, PH, SW, and VIs, such as NDVI and PRI, which is consistent with the results of previous studies [61,62]. Similar results have previously been reported for numerous crops, such as barley [63], wheat *(Triticum aestivum*) [64,65,66], maize [67,68,69], sugarcane *(Saccharum officinarum)* [70], cucumber [71], tomato *(Solanum lycopersicum)* [72], and Tef *(Eragrostis tef* (Zucc.) Trotter) [73]. The observed improvements in the shoot characteristics potentially generated beneficial effects on plant growth and development by boosting chlorophyll levels, enhancing photosynthetic plant capacity, and improving the overall growth and nutritional status, which eventually contributed to improved seed production or yield. The significant changes in these traits could likely be attributed to Si deposition that strengthens the cell walls, thereby enabling plants to resist biotic and abiotic stresses, such as wind and pest invasion [74]. A positive association between PH and Chl with PN and TGN further confirmed the assumption. Similar findings, demonstrating improvements in shoot and physiological characteristics leading to increased crop yield, have previously been reported [1,73,75]. Plant roots are fundamental structures for plant growth and development and are crucial in determining crop yield, primarily due to their ability to absorb inorganic minerals and water from the soil, which are essential for plant metabolic processes [76]. A comprehensive understanding of root traits, such as root length, root angle, root density, and root diameter, is crucial for improving crop output. The abovementioned root morphological characteristics are critical in determining the nutrient and water absorption capacity of plants from the soil as well as resistance to various environmental stresses.

Our study revealed significant changes in root morphological traits in response to Si application. Notably, the positive effects of Si on various root morphological features, including TRL, SA, RV, NT, and NF, were observed after treatment. AD, which is negatively correlated with TRL, was reduced in Si-treated plants, suggesting that AD is reduced as the roots get finer or thinner. Moreover, these results were consistent across the two tested environments and with different treatments, indicating the stable impact of Si application on root morphology. Similarly, a previous study demonstrated the effects of Si on the morphology of date palm roots (*Phoenix dactylifera*) [77]. The improved TRL and SA observed in this study are potentially significant for enhancing crop nutritional status and yield. An increase in the length and SA of the root system facilitates the accessibility of plants to essential nutrients from the soil, which can promote overall crop growth and development as well as yield. In the present study, a positive correlation was observed between TRL and key yield traits, such as PN and TGN, suggesting that improvement in root morphology, particularly TRL and SA, could profoundly impact plant growth and productivity, underscoring the importance of considering root traits in the efforts to enhance crop yields. Our results also indicated that Si application could significantly increase crop yield, which is consistent with that observed in other plants [73,78,79,80,81]. Notably, varied responses to different Si concentrations were detected, indicating crop and location-specific requirements for Si. For example, T2 treatment (4.6 kg Si per plot) was found to be more effective in both environments. At Gyeongsan (35°48′01.9″ N 128°53′1″ E), T1 (2.3 kg Si per plot)-treated plants showed a significant increase of 10.9% in their yield, whereas T2 (4.6 kg Si per plot)-treated plants demonstrated a significant increase of 22.7% in their yield, indicating a difference of 10.5% in yield between the T1 and T2 treatments. At Gunwi (36°06′37.0″ N 128°38′42.9″ E), T1-treated plants showed a 14.2% increase in yield compared with the control, whereas T2 showed a 25.6% increase in yield, indicating a 9.9% difference in the output between the two treatments. The difference in yield output can be attributed to the variability in the soil properties. The pH of the soil at Gyeongsan and Gunwi differed slightly (6.2 and 6.0, respectively) (Appendix A), whereas other important properties, such as cation exchange capacity (190 and 158 mmol (c)/kg) and total nitrogen (279 and 299 cg/kg) showed considerable differences. In addition, the soil composition demonstrated differences, with variations in the clay, sand, and silt percentage (Appendix A); these variations, along with Si fertilizer, may have probably led to the differences in the yield output. Our study suggests that the optimum concentration of Si, knowledge of crop species, and the nutritional status of the cultivated area need to be considered while choosing the right dose of fertilizers. In addition, a genomic-level study is crucial to determine the mechanism of Si transport and its role in modifying the overall morphological and physiological traits of plants.

## 4. Materials and Methods

### 4.1. Land Suitability Analysis

For our experiment, two different locations in the Republic of South Korea were selected, namely, Gyeongsan (35°48′01.9″ N 128°53′1″ E) and Gunwi (36°06′37.0″ N 128°38′42.9″ E). QGIS version 3.28.1, a piece of open-source geographic information system software, was used to perform the land suitability analysis and predict the appropriateness of land for soybean production. This involved defining the two selected study sites and inputting the raster layers of soil pH, nitrogen, organic carbon, cation exchange capacity, and erosion potential from SoilGrids (ISRIC, 2020, Version 2.0) [82], as well as the rainfall data from Giovanni spatial databases [83]. The Ecocrop model was used to assign weights and critical values to each layer based on their relative importance to production (FAO, 2022) [84]. The 1976 FAO land suitability classification system [85] was used to rank the suitability of each location. The results revealed that the selected sites were within the moderately suitable class, which indicated their potential for soybean production (Figure 7).

### 4.2. Experimental Design and Treatments

This study was conducted during the growing season of the year 2022 at two different sites, i.e., the Kyungpook National University Research Farm, Gunwi, and the Gyeongsan Research Farm, Republic of Korea. A widely cultivated soybean variety, Daechan, was used in the experiment. The seeds were directly sown in the field on 22 May 2022 and 3 June 2022 using a hand planter (TR110RA, Agritecno Yazaki Korea, Suwon City, Republic of Korea), with a sowing rate of two seeds per 10 cm. A randomized complete block design was set with three replications and three treatments, including a control and two different Si fertilizer amounts (dose) of 2.3 and 4.6 kg per plot, representing treatment T1 and treatment T2, respectively. The dimensions of each main plot were 9 m × 9 m, whereas the dimensions of the research plot were 5 m × 2 m, with a 1 m gap between each of the four rows. Before seeding, ridges measuring 0.3 m × 0.7 m (height × width) were constructed in each plot. Commercial silt fertilizer with a ratio of 40% CaO, 2% MgO, and 25% SiO_2_ (Jecheon Ceramic Co., Ltd., Busan, Republic of Korea) was used. The Si fertilizer treatments were applied a day before sowing using the broadcasting method, whereas the control was untreated. Furthermore, no additional fertilizers were added. A commercial herbicide containing the active ingredient glufosinate-ammonium 18% (local name: Golddara, Syngenta, Basel, Switzerland) and an insecticide containing Etofenprox (10%) as an active ingredient (local name: Myeong tazaja, Dongbu Farm Hannong, Seoul, Republic of Korea) were applied at a concentration of 60 mL/20 L and 20 mL/20 L of water, respectively. The herbicide and insecticide were applied twice, once at the time of planting and second at the V4 vegetative growth stage. Data on agronomic traits were collected during the 2022 growing season. During May–October 2022, Gyeongsan had a cumulative rainfall of 707 mm and an average monthly temperature ranging from 14 °C to 25 °C, whereas Gunwi experienced a total rainfall of 805 mm and an average monthly temperature ranging from 13 °C to 25 °C (Beaudoing, H. and M. Rodell, [86] NASA/GSFC/HSL (2020), GLDAS Noah Land Surface Model L4 monthly 0.25 × 0.25 degree V2.1, Greenbelt, MD, USA, Goddard Earth Sciences Data and Information Services Center (GES DISC), Accessed: 25 October 2022, https://doi.org/10.5067/SXAVCZFAQLNO). Detailed information about the soil properties and weather conditions are given in Appendix A.

### 4.3. Measurement of Agronomic Traits

#### 4.3.1. Measurement of Shoot Characteristics (PH, Stem Width [SW], and the Number of Lateral Branches)

Data were collected at three time points, beginning from 64 DAP (vegetative stage: V5) and then at 14-day intervals (reproductive stage: R1 to R2 and R2 to R3, respectively). Two rows of 15 plants per replication were selected. PH and SW were measured using a ruler and vernier caliper, respectively, whereas the NB per plant was manually counted.

#### 4.3.2. Measurement of Photosynthetic Parameters (Vegetative Indices and Chlorophyll Content)

Photosynthetic parameters, such as chlorophyll content (Chl) and leaf VIs, including NDVI and PRI, were measured at 2-week intervals, with the former being a highly effective VI for quantifying green vegetation. The NDVI normalizes green leaf scattering in the near-infrared wavelength and chlorophyll absorption in the red wavelength. PRI is responsible for changes in carotenoid pigments, specifically xanthophylls pigments (yellows) that are absorbed by live foliage. These pigments signify photosynthetic light use efficiency and are useful to quantify vegetative production and stress levels. Photosynthesis-related parameters were analyzed at 64, 79, and 93 DAP at both experimental locations. A chlorophyll meter (MC-100, Apogee Instruments Inc., Logan, UT, USA) was used to collect uniform data on chlorophyll content and VIs from a selected trifoliate leaf from the stem. The chlorophyll content and VIs were measured with PolyPen (RP410, Photon Systems Instruments, Brno, Czech Republic). All data were collected in triplicates.

#### 4.3.3. Determination of Soybean Root and Yield Traits

Full mature soybean plants at reproductive stage 8 (R8), when 95% of the pods have reached their mature pod color, were harvested from Gunwi and Gyeongsan on 12 and 13 October 2022, respectively. To reduce the border effects, shoots from the two middle rows within the plots were picked, leaving three rows between each treatment. Therefore, the total size of the seed collection area was 10 m^2^ per plot. The shoots were cut using a sickle, placed in a mesh bag, and allowed to dry in a greenhouse before threshing. Root samples were collected after harvesting the shoots. The roots were removed by demarcating a 30 cm diameter circle around the target plants. A shovel was then used to dig out 10 roots per treatment, making a total of 90 root samples per treatment for each location, which were used to acquire images.

#### 4.3.4. Analysis of Root Morphological Traits

The root morphological traits were analyzed by first removing all debris and soil particles from the roots. A rhizobox was set up on a mounted table for clear imaging. The root samples were hung inside the rhizobox, and root images were captured using a mirrorless RGB camera (Canon EOS M200, Tokyo, Japan) (Table 2), with an EF-M 15–45 mm lens. Root traits, such as projected area, length, and diameter, were analyzed using WinRHIZO software Pro (WinRHIZO, Regent Instruments Inc., Quebec, QC, Canada), and different root characteristics, including TRL, SA, AD, NT, RV, and NF, were measured. 

#### 4.3.5. Statistical Analysis

The experiment was set in a randomized design with three replications. To determine statistical significance, repeated ANOVA measures were conducted using SAS (9.4; SAS Gary, NC, USA) and the SciPy library in Python. Duncan’s multiple range test was performed to compare the means between treatments at *p* ≤ 0.05, *p* ≤ 0.01, and *p* ≤0.0001. Graphical visualization was performed in Python (seaborn and matplotlib library) and Microsoft Excel.

## 5. Conclusions

In summary, our results demonstrated that Si fertilizer application positively impacted the growth and development of soybean plants by modifying both the shoot and root morphological traits, thereby leading to higher yields. Our findings add to the growing body of literature that highlights the beneficial effects of Si fertilizer on crop yield and sustainable production. Moreover, an analysis of the effects of different Si treatments on crop output in this study identified the location-specific differences in total yield, with T2 (22.8% and 25.6%, representing an output of 2.19 and 2.24 t ha^−1^ at Gyeongsan and Gunwi, respectively) and T1 (11% and 14.2%, representing an output of 1.98 and 2.04 t ha^−1^ at Gyeongsan and Gunwi, respectively) proving its potential for enhancing crop output and promoting sustainable production. These findings will facilitate future in-depth studies on the effectiveness of Si fertilizer in crop production.

## Figures and Tables

**Figure 1 plants-12-02190-f001:**
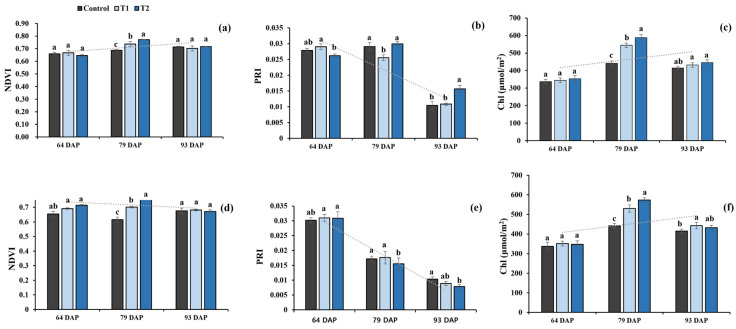
Effects of Si fertilizer treatment on different vegetative indices at the Gyeongsan and Gunwi experimental locations. (**a**) NDVI, Gyeongsan; (**b**) PRI, Gyeongsan; (**c**) Chlorophyll content, Gyeongsan; (**d**) NDVI, Gunwi (**e**) PRI, Gunwi, and (**f**) Chlorophyll content, Gunwi. Different lowercase letters above the error bar indicate significant differences at *p* ≤ 0.05; values are mean ± SE (*n* = 10).

**Figure 2 plants-12-02190-f002:**
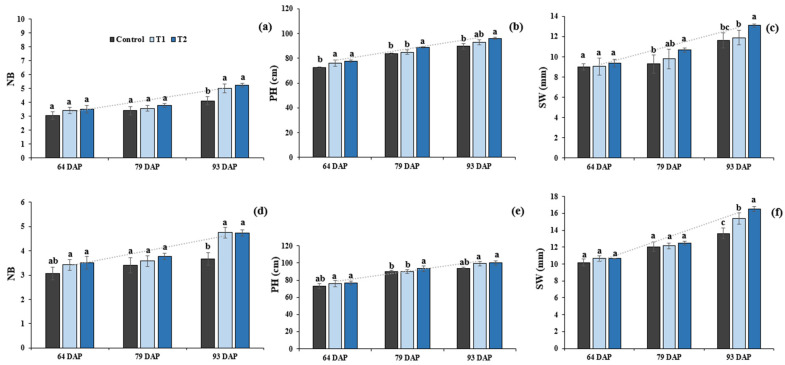
Effects of Si fertilizer treatment on different shoot traits at the Gyeongsan and Gunwi locations. (**a**) NB, number of branches, Gyeongsan; (**b**) PH, plant height, Gyeongsan; (**c**) SW, stem width, Gyeongsan; (**d**) NB, number of branches, Gunwi; (**e**) PH, plant height, Gunwi; and (**f**) SW, stem width, Gunwi. Different lowercase letters above the error bar indicate significant differences at *p* ≤ 0.05; values are mean ± SE (*n* = 10).

**Figure 3 plants-12-02190-f003:**
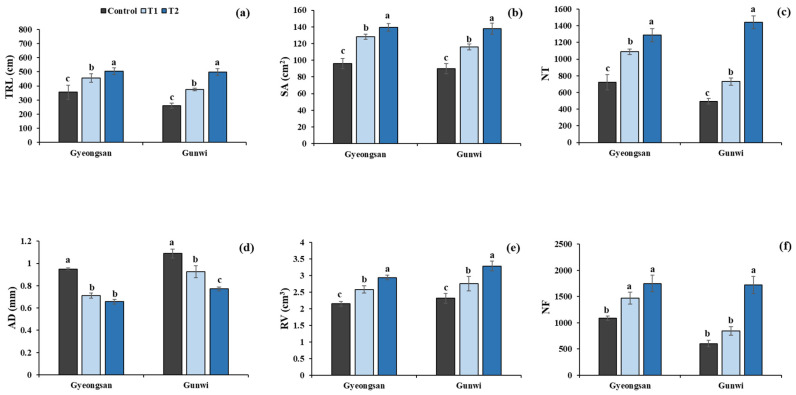
Effects of Si fertilizer treatment on different root traits at the Gyeongsan and Gunwi locations. (**a**) TRL, total root length; (**b**) SA, surface area; (**c**) NT, number of tips; (**d**) AD, average diameter; (**e**) RV, root volume, and (**f**) NF, number of forks. Different lowercase letters above the error bar indicate significant differences at *p* ≤ 0.05; values are mean ± SE (*n* = 10).

**Figure 4 plants-12-02190-f004:**
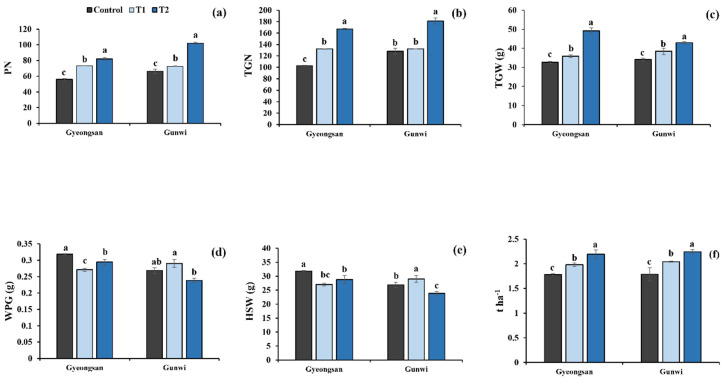
Effects of Si fertilizer treatment on different yield traits and total yield at the Gyeongsan and Gunwi locations. (**a**) PN, pod number; (**b**) TGN, total grains number; (**c**) TGW, total grains weight; (**d**) WPG, weight per grain; (**e**) HSW, hundred seeds weight; and (**f**) Yield, tons per hectare. Different lowercase letters above the error bar indicate a significant difference at *p* ≤ 0.05; values are mean ± SE (*n* = 10).

**Figure 5 plants-12-02190-f005:**
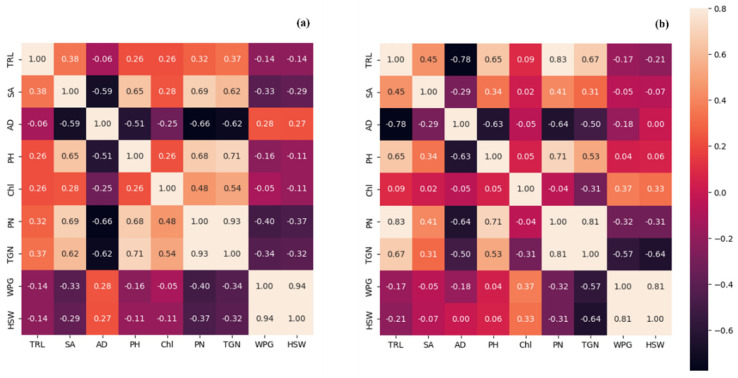
Pearson correlation coefficients of the key root, shoot, and yield traits were evaluated at (**a**) Gyeongsan and (**b**) Gunwi. Traits: TRL, total root length, SA: surface area, AD: average diameter, PH: plant height, Chl: chlorophyll content, PN: pod number, TGN: total grain number, WPG: weight per grain, and HSW: hundred seed weight. The heat maps are annotated with the correlation values (*r^2^)* shown with specific colors. The values assigned to each color are represented as a bar to the right of the figure with min–max (−0.6–0.8).

**Figure 6 plants-12-02190-f006:**
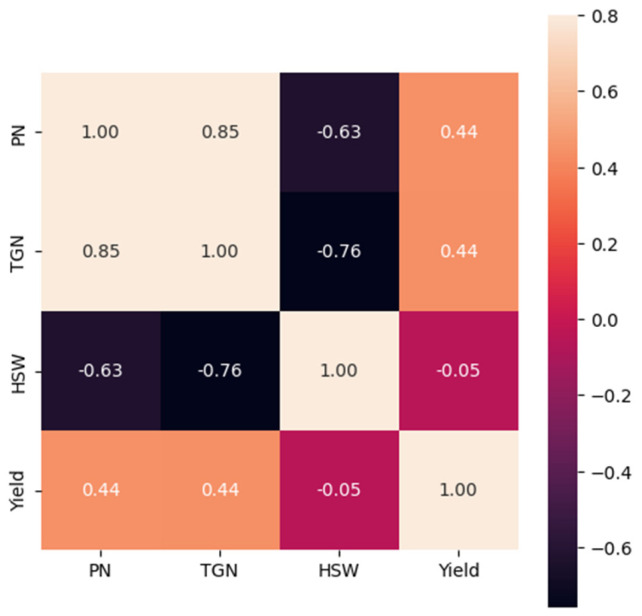
Pearson correlation coefficients of important yield traits and total yield of the two experimental locations. Traits: PN, pod number; TGN, total grain number; WPG, weight per grain; and HSW, hundred seed weight. The heat maps are annotated with correlation values (*r^2^)* shown with specific colors. The values assigned to each color are represented as a bar to the right of the figure with min–max (−0.6–0.8).

**Figure 7 plants-12-02190-f007:**
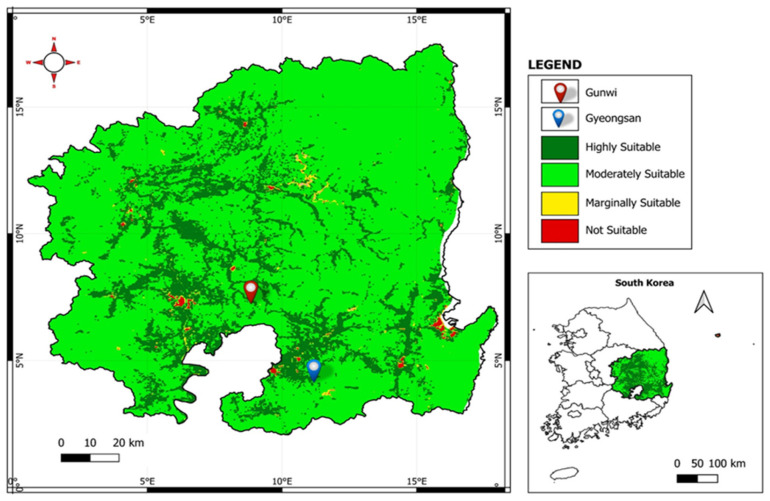
Land suitability map for soybean cultivation and production based on FAO standards [85]. The experimental sites were Gyeongsan (35°48′01.9″ N 128°53′1″ E) and Gunwi (36°06′37.0″ N 128°38′42.9″ E).

**Table 1 plants-12-02190-t001:** The results of the analysis of variance (ANOVA) of shoot, root, and yield traits.

Traits	Source	DF	Type III SS	Mean Square	F Value	*p* > F
Chl	Tre	2	191,537.704	95,768.852	18.34	<0.0001
	Rep	2	8088.87	4044.435	0.77	0.4615
	DC	2	2,790,622.344	1,395,311.172	267.18	<0.0001
	Env	2	150,359.098	75,179.549	14.4	<0.0001
	Tre*Env	2	19,221.88	9610.94	1.84	0.1598
NDVI	Tre	2	0.2027455	0.10137275	9.32	0.0001
	Rep	2	0.02557458	0.01278729	1.18	0.3093
	DC	2	0.19527912	0.09763956	8.98	0.0001
	Env	1	0.0924627	0.0924627	8.5	0.0037
	Tre*Env	2	0.08579846	0.04289923	3.94	0.0199
PRI	Tre	2	0.00023445	0.00011722	0.69	0.5018
	Rep	2	0.0007302	0.0003651	2.15	0.1174
	DC	2	0.02506911	0.01253455	73.83	<0.0001
	Env	1	0.00225234	0.00225234	13.27	0.0003
	Tre*Env	2	0.00028028	0.00014014	0.83	0.4386
PH	Tre	2	2868.03333	1434.01667	50.82	<0.0001
	Rep	2	846.34444	423.17222	15	<0.0001
	DC	2	37,379.74444	18,689.87222	662.32	<0.0001
	Env	1	1833.37963	1833.37963	64.97	<0.0001
	Tre*Env	2	41.4037	20.70185	0.73	0.4807
SW	Tre	2	95.763693	47.881847	9.25	0.0001
	Rep	2	51.570641	25.785321	4.98	0.0072
	DC	2	1358.75383	679.376915	131.18	<0.0001
	Env	1	595.413002	595.413002	114.97	<0.0001
	Tre*Env	2	8.454973	4.227487	0.82	0.4426
NB	Tre	2	41.7148148	20.8574074	11.94	<0.0001
	Rep	2	0.337037	0.1685185	0.1	0.908
	DC	2	523.937037	261.9685185	149.99	<0.0001
	Env	1	63.3796296	63.3796296	36.29	<0.0001
	Tre*Env	2	0.6259259	0.312963	0.18	0.836
TRL	Tre	2	1,208,726	604,363.2	40.51	<0.0001
	Rep	2	37,639.47	18,819.73	1.26	0.2858
	Env	1	138,238.1	138,238.1	9.27	0.0027
	Tre*Env	2	77,379.17	38,689.58	2.59	0.0777
SA	Tre	2	70,151.92	35,075.96	56.26	<0.0001
	Rep	2	3276.528	1638.264	2.63	0.0751
	Env	1	2781.931	2781.931	4.46	0.0361
	Tre*Env	2	457.2975	228.6487	0.37	0.6935
AD	Tre	2	4.478359	2.239179	59.67	<0.0001
	Rep	2	0.113299	0.05665	1.51	0.2239
	Env	1	1.410859	1.410859	37.6	<0.0001
	Tre*Env	2	0.147526	0.073763	1.97	0.1432
RV	Tre	2	25.03816	12.51908	17.8	<0.0001
	Rep	2	0.120214	0.060107	0.09	0.9181
	Env	1	2.811756	2.811756	4	0.0471
	Tre*Env	2	0.362826	0.181413	0.26	0.7729
NT	Tre	2	17,371,086	8,685,543	111.23	<0.0001
	Rep	2	231,728.6	115,864.3	1.48	0.2296
	Env	1	918,502.5	918,502.5	11.76	0.0008
	Tre*Env	2	1,912,638	956,318.9	12.25	<0.0001
NF	Tre	2	25,309,441	12,654,721	35.68	<0.0001
	Rep	2	986,063.9	493,032	1.39	0.2518
	Env	1	6,172,119	6,172,119	17.4	<0.0001
	Tre*Env	2	3,318,264	1,659,132	4.68	0.0105
HSW	Tre	2	186.2892	93.1446	19.7	<0.0001
	Rep	2	0.15731	0.078655	0.02	0.9835
	Env	1	613.5846	613.5846	129.78	<0.0001
	Tre*Env	2	611.4961	305.7481	64.67	<0.0001
WPG	Tre	2	0.01684	0.00842	18.53	<0.0001
	Rep	2	0.001639	0.000819	1.8	0.1679
	Env	1	0.050501	0.050501	111.12	<0.0001
	Tre*Env	2	0.06271	0.031355	68.99	<0.0001
TGW	Tre	2	6486.6	3243.3	535.45	<0.0001
	Rep	2	14.55694	7.278472	1.2	0.3032
	Env	1	136.4335	136.4335	22.52	<0.0001
	Tre*Env	2	1696.049	848.0246	140	<0.0001
TGN	Tre	2	107,212.3	53,606.16	2359.42	<0.0001
	Rep	2	15.8111	7.9056	0.35	0.7066
	Env	1	11,139.2	11,139.2	490.28	<0.0001
	Tre*Env	2	5703.6	2851.8	125.52	<0.0001
PN	Tre	2	27,033.3	13,516.65	1159.01	<0.0001
	Rep	2	34.23333	17.11667	1.47	0.2333
	Env	1	4460.089	4460.089	382.44	<0.0001
	Tre*Env	2	2826.678	1413.339	121.19	<0.0001
Yield	Tr	2	0.562403	0.281202	16.11	0.0007
	Rep	2	0.005328	0.002664	0.15	0.8604
	Env	1	0.006393	0.006393	0.37	0.5585
	Tr*Env	2	0.002967	0.001484	0.09	0.919

Tre, treatment; Rep, replication; Env, environment; Tre*Env, treatment × environment interaction; DC, data collection; Chl, chlorophyll; NDVI, normalized difference vegetation index; PRI, photochemical reflectance index; PH, plant height; SW, stem width; NB, number of branches; TRL, total root length; SA, surface area; AD, average diameter; RV, root volume; NT, number of tips; NF, number of forks; HSW, hundred seed weight; WPG, weight per grain; TGW, total grain weight; TGN, total grain number; PN, pod number.

**Table 2 plants-12-02190-t002:** Camera specifications.

Feature	Specification
Product dimensions	109.22 × 66.04 × 35.56 mm
Weight	539.78 g
Image sensor	24.1 Megapixel CMOS (APS-C)
Image processor	DIGIC 8
Lens	EF-M 15–45 mm IS STM

## Data Availability

Not applicable.

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
