# Peer review of "Evaluating the Effectiveness of Calcium Silicate in Enhancing Soybean Growth and Yield"

_plants, 2023, doi:10.3390/plants12112190_

Round 1

Reviewer 1 Report

This study reports the results of the experiment conducted in the field to evaluate the silicon effect on growth and yield traits of soybean. In general, this manuscript addresses an interesting and important topic. Nonetheless, a major revision is needed before the manuscript is considered to be published in this journal. Some specific comments are mentioned below:

Keywords:

L36: too many keywords. For example, choose between ‘yield’ and ‘yield traits’.

Avoid suitibility analysis.

Introduction

L40: glycine in italic and avoid the 2 comma.

L59: sugarcane (Saccharum officinarum) [12], rice (the comma).

L61-91: there is some redundancy regarding the effect of Si on plants.

Results: you need to focus on the most relevant results. too much detail in the description of the results.

L98 and L231: delete the point.

L105-111: rephrase.

Figure 1, 2, 3 and 4: the figures are not readable. Please, increase the size.

L252: (Zucc.) Trotter) without italic.

Materials and methods

L343: you should indicate that the mode of application is fertigation.

L347: the fertilizer does not contain only Si but also Ca and Mg so the observed effect is not due to Si only. Or, there are 2 fertilizers used. Specify.

L353-355: already mentioned in the following section.

L357: Avoid the abbreviations in the titles.

L358: give more details about fertilization. What type of Si fertilizer (the chemical formula).

L395: mention traits in full and abbreviations in parentheses.

This study reports the results of the experiment conducted in the field to evaluate the silicon effect on growth and yield traits of soybean. In general, this manuscript addresses an interesting and important topic. Nonetheless, a major revision is needed before the manuscript is considered to be published in this journal. Some specific comments are mentioned below:

Keywords:

L36: too many keywords. For example, choose between ‘yield’ and ‘yield traits’.

Avoid suitibility analysis.

Introduction

L40: glycine in italic and avoid the 2 comma.

L59: sugarcane (Saccharum officinarum) [12], rice (the comma).

L61-91: there is some redundancy regarding the effect of Si on plants.

Results: you need to focus on the most relevant results. too much detail in the description of the results.

L98 and L231: delete the point.

L105-111: rephrase.

Figure 1, 2, 3 and 4: the figures are not readable. Please, increase the size.

L252: (Zucc.) Trotter) without italic.

Materials and methods

L343: you should indicate that the mode of application is fertigation.

L347: the fertilizer does not contain only Si but also Ca and Mg so the observed effect is not due to Si only. Or, there are 2 fertilizers used. Specify.

L353-355: already mentioned in the following section.

L357: Avoid the abbreviations in the titles.

L358: give more details about fertilization. What type of Si fertilizer (the chemical formula).

L395: mention traits in full and abbreviations in parentheses.

Author Response

Author's Reply to the Review Report (Reviewer 1)

Comments and Suggestions for Authors

This study reports the results of the experiment conducted in the field to evaluate the silicon effect on growth and yield traits of soybean. In general, this manuscript addresses an interesting and important topic. Nonetheless, a major revision is needed before the manuscript is considered to be published in this journal. Some specific comments are mentioned below:

Answer: We would like to thank the worthy reviewer for the time given to this manuscript. All the comments and suggestions were genuine, and valuable improved the quality of the manuscript. We highly appreciate your efforts and agree to the suggested changes. All the changes made in the revised MS can be found with the track change, highlighted in green color.

Keywords:

L36: too many keywords. For example, choose between ‘yield’ and ‘yield traits’.

Avoid suitibility analysis.

Answer: Thank you for the suggestion, we have incorporated the suggested changes.

Introduction

L40: glycine in italic and avoid the 2 comma.

L59: sugarcane (Saccharum officinarum) [12], rice (the comma).

L61-91: there is some redundancy regarding the effect of Si on plants.

Answer: Thank you for the suggestion, we have corrected the mistakes and removed the redundant content.

Results: you need to focus on the most relevant results. too much detail in the description of the results.

L98 and L231: delete the point.

L105-111: rephrase.

Figure 1, 2, 3 and 4: the figures are not readable. Please, increase the size.

L252: (Zucc.) Trotter) without italic.

Answer: Thank you for the valuable suggestion, we have corrected the mistakes and incorporated the suggested changes.

Materials and methods

L343: you should indicate that the mode of application is fertigation.

Answer: Thank you for the valuable suggestion, we have provided the information according to the suggestion.

L347: the fertilizer does not contain only Si but also Ca and Mg so the observed effect is not due to Si only. Or, there are 2 fertilizers used. Specify.

Answer: We agree that fertilizer does contain Ca, and Mg, but the fertilizer is commonly known as (super silicic acid) and it is commonly known that the application of silt fertilizer or silicate fertilizer increased soil pH, due it enhances the exchangeable of Ca, Mg and other mineral salts as well as aid in increasing phosphate and moisture contents in soil. So, its Si which facilitates this process and secondly molar mass of SiO2 is higher (Molar mass of CaO: 56.0774 g/mol; MgO: 40.3044 g/mol; and SiO2: 60.08 g/mol) that’s why the majority of studies though used the potassium silicate or calcium silicate they highlighted the effect of Si and not the mineral salts (Ca and Mg).

L353-355: already mentioned in the following section.

Answer: Thank you for the suggestion, we have removed the content as per the suggestion.

L357: Avoid the abbreviations in the titles.

L358: give more details about fertilization. What type of Si fertilizer (the chemical formula).

L395: mention traits in full and abbreviations in parentheses.

Answer: Thank you for the suggestion, we have made the suggested changes we have already provided the trait’s full name at first instance and then abbreviated throughout the MS.

Reviewer 2 Report

Comments and suggestions for authors

General comments

The manuscript presents research results on the application effectiveness of Si fertilizers (Ca2SiO4) on the rate of vegetative growth and of certain physiological and productivity indices in soybeans in two different locations.

In principle, the purpose of this study was to contribute to a better understanding of the physiological mechanisms, involved in the Si absorption, translocation, and its implication in different metabolic pathways with a role in growth and productivity in Glycine max.

The experimental way is interesting, based on a multitude of determinations and analysis like; growth dynamic of shoots and roots (plants height, stem width, number of branches, total root length and root volume), physiological parameters (chlorophyll content in foliar apparatus, photochemical reflectance index) and productivity indicators (hundred seed weight, weight per grain, total grain weight,  total grain number and pod number).

Some of the used methods are correct but simplistic, based mainly on measurement, counting, and weighing, and another category of techniques is based on the use of determination equipment with limited accuracy (chlorophyll meters). Hence, there are still many aspects that need to be corrected and /or completed.

Title

Evaluating the Effectiveness of Silicon Fertilizer in Enhancing  Soybean Growth and Yield: A Comparative Study. – the title can be improved, a single fertilizer containing Si was used and it is not even a comparative study...

Abstract

It must be corrected to contain essential information.

Lines 17-21. Not required, this information can be moved to the introduction…

Lines 23-25. No description of the software used for geographic information system.

Keywords

Too many keywords, some are also found in the title - need to be corrected.

Introduction

It needs to be reviewed, much more focused on the current state of knowledge on the impact of the application of Si on metabolism in soya and legumes… and highlighted the effects on growth, development, and productive plants capacity.

Results

It is unfortunate to place table 1 with the results on the variance analysis at the beginning, because it is a synthetic table that brings together the recorded results of all the studied parameters... With a large volume of data, it must also be commented on...

Fig 1, 2, 3 and 4 are too small, the data cannot be properly analyzed... Comments refer only to percentage values, instead of commenting and analyzing the actual values specific to each parameter.

Correlations between yield and root shoot traits are misinterpreted, some values presented do not indicate the existence of any positive or negative correlation...

Discussions

are too general, they must be focused on the results of the application of Si to soybeans, highlighting the impact on the architecture and growth of vegetative organs, productivity, the promotion of nodules formation and function, and protection from biotic and abiotic stressors. The literature from the main flow of information abounds with results on the mentioned issues. Therefore, this chapter needs a drastic correction and re-approach.

Material and methods

Apart from the geographical location, no data are presented on soil and climate conditions, not even in supplementary files…

It is not clear what are the amounts of Si-based fertilizers, calcium orthosilicates (Ca2SiO4) and metasilicate calcium (CaSiO3) applied in each variant.

The time of carrying out the determinations expressed in DAP does not show the stage of development of the plants, usually a development code is used (ex. BBCH).

The methods for determining photosynthetic parameters by using the specified equipment are estimated not analytical... I do not recommend using them in in-depth research.

The manuscript needs a detailed analysis and improvements, because in its current form it cannot be published.

Good luck!

I recommend a future resubmission.

Author Response

Author's Reply to the Review Report (Reviewer 2)

Comments and suggestions for authors

General comments

The manuscript presents research results on the application effectiveness of Si fertilizers (Ca2SiO4) on the rate of vegetative growth and of certain physiological and productivity indices in soybeans in two different locations. In principle, the purpose of this study was to contribute to a better understanding of the physiological mechanisms, involved in the Si absorption, translocation, and its implication in different metabolic pathways with a role in growth and productivity in Glycine max.

The experimental way is interesting, based on a multitude of determinations and analysis like; growth dynamic of shoots and roots (plants height, stem width, number of branches, total root length and root volume), physiological parameters (chlorophyll content in foliar apparatus, photochemical reflectance index) and productivity indicators (hundred seed weight, weight per grain, total grain weight,  total grain number and pod number).

Some of the used methods are correct but simplistic, based mainly on measurement, counting, and weighing, and another category of techniques is based on the use of determination equipment with limited accuracy (chlorophyll meters). Hence, there are still many aspects that need to be corrected and /or completed.

Answer: We would like to thank the worthy reviewer for the time given to this manuscript. All the comments and suggestions were genuine, and valuable improved the quality of the manuscript. We highly appreciate your efforts and agree to the suggested changes. All the changes made in the revised MS can be found with the track change, highlighted in green color.

Title

Evaluating the Effectiveness of Silicon Fertilizer in Enhancing  Soybean Growth and Yield: A Comparative Study. – the title can be improved, a single fertilizer containing Si was used and it is not even a comparative study.

Answer: Thank you for the suggestion, comparative in a sense we used it because the study was conducted at two different geographic locations, however, we agree with the worthy reviewer and thus we have changed the Title. Once again thank you very much for your valuable suggestions.

Abstract

It must be corrected to contain essential information.

Lines 17-21. Not required, this information can be moved to the introduction…

Lines 23-25. No description of the software used for geographic information system.

Answer: Thank you for the suggestion, we have made the suggested changes.

Keywords

Too many keywords, some are also found in the title - need to be corrected.

Answer: Thank you for the suggestion, we have removed excess keywords.

Introduction

It needs to be reviewed, much more focused on the current state of knowledge on the impact of the application of Si on metabolism in soya and legumes… and highlighted the effects on growth, development, and productive plants capacity.

Answer: Thank you for the suggestion, we have revised the introduction content and added the relevant studies reported on soybean and legumes hope it solves the purpose.

Results

It is unfortunate to place table 1 with the results on the variance analysis at the beginning, because it is a synthetic table that brings together the recorded results of all the studied parameters... With a large volume of data, it must also be commented on...

Answer: Thank you for the suggestion, but there are several studies and even our own group published several articles with similar content in the results section. However, we have modified the results and removed too minute details provided earlier, keeping the other reviewer’s comment in mind.

Fig 1, 2, 3 and 4 are too small, the data cannot be properly analyzed... Comments refer only to percentage values, instead of commenting and analyzing the actual values specific to each parameter.

Answer: Thank you for the suggestion, as per the suggestions we have increased the Figure’s font and size, however, we do not find there is any wrong in the data analysis as we used the best suitable statistical test for the analysis of these data and results interpreted based on the analyzed raw data. Mentioning raw data for each parameter would not be the appropriate method and previous several studies, including our own studies published in reputed journals, followed a similar representation of data as we have provided in this MS. Hence, we would like to request the reviewer to reconsider the view.

Correlations between yield and root shoot traits are misinterpreted, some values presented do not indicate the existence of any positive or negative correlation...

Answer: Thank you for the suggestion, as per the suggestions we cross-checked and also removed mentioned R2 and minute details as suggested by other reviewers. We just illustrated significant correlations (positive or negative) in the results.

Discussions

are too general, they must be focused on the results of the application of Si to soybeans, highlighting the impact on the architecture and growth of vegetative organs, productivity, the promotion of nodules formation and function, and protection from biotic and abiotic stressors. The literature from the main flow of information abounds with results on the mentioned issues. Therefore, this chapter needs a drastic correction and re-approach.

Answer: Thank you for commenting on it we have made the suggested changes and just highlighted the studies with the Si effect on plant growth and productivity in the discussion.

Material and methods

Apart from the geographical location, no data are presented on soil and climate conditions, not even in supplementary files…

Answer: Sorry for the inconvenience, as per the valuable suggestions we have now provided the additional information in the supplementary file (Table S1).

It is not clear what are the amounts of Si-based fertilizers, calcium orthosilicates (Ca2SiO4) and metasilicate calcium (CaSiO3) applied in each variant.

Answer: Thank you for commenting on it, but we have clearly provided the information as  “A randomized complete block design was set with three replications and three treatments, including a control and two different Si fertilizer amounts (dose) of 2.3 and 4.6 kg per plot, representing treatment T1 and treatment T2, respectively. Commercial silt fertilizer with a ratio of 40% CaO, 2% MgO, and 25% SiO2 (Jecheon Ceramic Co., Ltd., South Korea) was used.”

The time of carrying out the determinations expressed in DAP does not show the stage of development of the plants, usually a development code is used (ex. BBCH).

Answer: Sorry for the inconvenience now we have provided the details in the revised MS about the stage “Data were collected at three time points, beginning from 64 DAP (vegetative stage: V5) and then 14 days intervals (reproductive stage: R1 to R2 and R2 to R3, respectively)”.

The methods for determining photosynthetic parameters by using the specified equipment are estimated not analytical... I do not recommend using them in in-depth research.

Answer: Thank you very much for your kind suggestions, we will avoid using specific equipment in our future research.

The manuscript needs a detailed analysis and improvements because in its current form, it cannot be published.

Good luck!

I recommend a future resubmission.

Answer: We would like to thank the worthy reviewer for the time given to this manuscript. We have either provided the justification or incorporated/provided the required information, hope the current form MS is suitable for the publications.

Reviewer 3 Report

The manuscript entitled: "Evaluating the Effectiveness of Silicon Fertilizer in Enhancing Soybean Growth and Yield: A Comparative Study" is of interest to the science and practice of agriculture. I appreciate that this is a field experience, located in two locations. However, it is better to repeat the experiment the next year. Weather conditions often modify the obtained results. I recommend the manuscript for publication in the journal Plants. However, the text needs to be corrected. I included detailed comments in the original text (see pdf).

General notes:

Write in the Abstract and Conclusion what were the significant differences between the locations, e.g. yield
change the word "soybean health"
line 40, write the Latin name of soybeans in italics
Write a short research hypothesis
Check the statistical calculations in Figures 1 and 4
line 280 correct the reference to the bibliography
In Materials and Methods, write what was the forecrop, chemical protection of the plantation, sowing rate, NPK fertilization, seeds inoculated, etc.
Write weather and soil conditions for each location.
Explain all abbreviations when you use them for the first time
Insert a reference to table 2 into the text
Revise the bibliography as required by the journal

I hope that my comments will help to improve the text of the manuscript.y

Author Response

Author's Reply to the Review Report (Reviewer 3)

Comments and Suggestions for Authors

The manuscript entitled: "Evaluating the Effectiveness of Silicon Fertilizer in Enhancing Soybean Growth and Yield: A Comparative Study" is of interest to the science and practice of agriculture. I appreciate that this is a field experience, located in two locations. However, it is better to repeat the experiment the next year. Weather conditions often modify the obtained results. I recommend the manuscript for publication in the journal Plants. However, the text needs to be corrected. I included detailed comments in the original text (see pdf).

General notes:

Write in the Abstract and Conclusion what were the significant differences between the locations, e.g. yield
change the word "soybean health"
line 40, write the Latin name of soybeans in italics
Write a short research hypothesis
Check the statistical calculations in Figures 1 and 4
line 280 correct the reference to the bibliography
In Materials and Methods, write what was the forecrop, chemical protection of the plantation, sowing rate, NPK fertilization, seeds inoculated, etc.
Write weather and soil conditions for each location.
Explain all abbreviations when you use them for the first time
Insert a reference to table 2 into the text
Revise the bibliography as required by the journal

I hope that my comments will help to improve the text of the manuscript.y

Answer: We would like to express our gratitude to the deserving reviewer for their time. All of the honest and helpful remarks and ideas raised the manuscript's caliber. We are very grateful for your efforts and accept the suggested comments. The track change, which is indicated in green, contains all the modifications made to the amended MS.

Reviewer 4 Report

Review

Manuscript ID: plants-2359423

The Authors have investigated an interesting topic – effect of  silicon fertilizer on soybean growth and yield.  In general, the content and methods used are correct. Unfortunately, the manuscript is incomplete and some parts of the manuscript need to be corrected.

1)      I suggest using the full name of the fertilizer in the text - calcium silicate. Besides Si, the fertilizer also contains calcium (even more than Si). Therefore, it is worth mentioning this element in the discussion, as well as assessing its role in shaping the growth and yield of plants in the experiment.

2)      I suggest changing the order of the figures in subsection 2.2. , if the first characteristic to be discussed is the chlorophyll content. Currently, the authors start with the description of the results for Figure 1c, not Figure 1a.

3)      The methodology lacks information on the content of Si in the soil and the potential of the soil to supply plants with Si. The reaction depends on the Si content in the soil and other soil characteristics. Also missing from the discussion is a discussion of soil conditions and the effect of soil on the observed responses of soybeans to fertilization.

4)      In addition, other soil data are missing. The map (Figure 7) only shows the soil's potential for soybean production.

5)      I suggest changing the manuscript title to: Evaluating the Effectiveness of Calcium Silicate in Enhancing Soybean Growth and Yield: A Comparative Study

Minor Notes:

6)      Line 40, the Latin name of soybeans should be in italics.

7)      Line 99, 104, the symbol P should be written in italics

8)      Table 1, please use "P" instead of "Pr > F". The explanation of the Dc symbol is missing. On the other hand, please use the uniform factor abbreviation, eg DC (please compare line 101)

9)      Subsection 2.3. and 2.4. For better clarity of the text, please remove the values of the coefficients of determination, as the correlation coefficients are given in Figure 5 and Figure 6. Moreover, the authors describe correlation, not regression, so there is no need for this.

10)  Please, in Figure 7, better indicate the location of the farms where the research was conducted.

11)  References - the font should be corrected, and journal names should be abbreviated.

Author Response

Author's Reply to the Review Report (Reviewer 4)

Reviewer 4,

The Authors have investigated an interesting topic – the effect of silicon fertilizer on soybean growth and yield.  In general, the content and methods used are correct. Unfortunately, the manuscript is incomplete and some parts of the manuscript need to be corrected.

Answer: We would like to thank the worthy reviewer for the time given to this manuscript. All the comments and suggestions were genuine, and valuable improved the quality of the manuscript. We highly appreciate your efforts and agree to the suggested changes. All the changes made in the revised MS can be found with the track change, highlighted in green color.

1)      I suggest using the full name of the fertilizer in the text - calcium silicate. Besides Si, the fertilizer also contains calcium (even more than Si). Therefore, it is worth mentioning this element in the discussion, as well as assessing its role in shaping the growth and yield of plants in the experiment.

Answer: Thank you very much for your valuable suggestions we have incorporated calcium silicate in the title. We agree that fertilizer does contain Ca, and Mg, but the fertilizer is commonly known as (super silicic acid) and it is commonly known that the application of silt fertilizer or silicate fertilizer increased soil pH, due it enhances the exchangeable of Ca, Mg and other mineral salts as well as aid in increasing phosphate and moisture contents in soil. So, its Si which facilitates this process and secondly molar mass of SiO2 is higher (Molar mass of CaO: 56.0774 g/mol; MgO: 40.3044 g/mol; and SiO2: 60.08 g/mol) that’s why the majority of studies though used the potassium silicate or calcium silicate they highlighted the effect of Si and not the mineral salts (Ca and Mg).

2)      I suggest changing the order of the figures in subsection 2.2. , if the first characteristic to be discussed is the chlorophyll content. Currently, the authors start with the description of the results for Figure 1c, not Figure 1a.

Answer: Thank you very much for your kind suggestions, we have modified the arrangement of text according to the suggestion instate of changing the figure hope it solves the purpose.

3)      The methodology lacks information on the content of Si in the soil and the potential of the soil to supply plants with Si. The reaction depends on the Si content in the soil and other soil characteristics. Also missing from the discussion is a discussion of soil conditions and the effect of soil on the observed responses of soybeans to fertilization.

Answer: Thank you very much for your kind suggestions, we have now provided the additional information in the supplementary file (Table S1) about the soil’s chemical properties. The geographic region is not known for containing Si and secondly, the experiment conducted in the field where we applied the Si fertilizer and control conditions, we do not apply any fertilizer so it suggests that changes in the plant attributes and yield are due to the application of Si fertilizer. However, we totally agree with the worthy reviewer that the reaction depends on the Si content in the soil and other soil characteristics, it would have been ideal to analyze soil properties before and after the application of fertilizer. In our future research, we will conduct a detailed analysis of soil to see the Si fertilizer effect in depth.

4)      In addition, other soil data are missing. The map (Figure 7) only shows the soil's potential for soybean production.

Answer: Thank you very much for your kind suggestions, we have now provided the additional information in the supplementary file (Table S1) about the soil’s chemical properties.

5)      I suggest changing the manuscript title to: Evaluating the Effectiveness of Calcium Silicate in Enhancing Soybean Growth and Yield: A Comparative Study

Answer: Thank you very much for your suggestions we have modified the Title.

Minor Notes:

6)      Line 40, the Latin name of soybeans should be in italics.

Answer: Thank you very much for your suggestions we have corrected it according to your suggestions.

7)      Line 99, 104, the symbol P should be written in italics

Answer: Thank you very much for your suggestions we have corrected it according to your suggestions.

8)      Table 1, please use "P" instead of "Pr > F". The explanation of the Dc symbol is missing. On the other hand, please use the uniform factor abbreviation, eg DC (please compare line 101)

Answer: Thank you very much for your suggestions we have revised it according to your suggestions.

9)      Subsection 2.3. and 2.4. For better clarity of the text, please remove the values of the coefficients of determination, as the correlation coefficients are given in Figure 5 and Figure 6. Moreover, the authors describe correlation, not regression, so there is no need for this.

Answer: Thank you very much for your suggestions we have revised it according to your suggestions.

10)  Please, in Figure 7, better indicate the location of the farms where the research was conducted.

Answer: Thank you very much for your suggestions we have incorporated the suggested changes in the revised MS.

11)  References - the font should be corrected, and journal names should be abbreviated.

Answer: Thank you very much for your suggestions we have incorporated the suggested changes in the revised MS. However, the MDPI journal/production team edits all the references in the appropriate style prior to publication.

Round 2

Reviewer 1 Report

The entire manuscript needs to be revised.

The manuscript needs extensive revision for language, grammar, punctuation, scientific name of species ...and overall style. The present paper should be revised by a qualified native English-speaking.

Author Response

Thanks for your comments. Based on your suggestion, we re-submitted our manuscript to English Editing Service which company name is Enago. Thus, I am convinced that current manuscript was taken extensive revision from qualified native English-speaking company.

Reviewer 2 Report

Corrections and manuscript level are insufficient, the proposed recommendations have not been followed, there is no cover letter with specific responses to comments. I still believe that the manuscript is below the level of the journal and has major experimentation deficiencies such as:

ü  Lack of analysis on the Si content and other soil macro and microelements,

ü  Lack of assessment of the different climatic conditions of the two locations on the determined parameters,

ü  Impossible methods of determining the roots growth by in situ experiments and partially harvesting them,

ü  The determination of the quantity of chlorophyll with the chlorophyll meter is an estimative method,

ü  Morphological determinations used by measurement and weighing... ???

ü  There are not used modern experimental techniques, biochemical analysis to be able to quantify the impact of the application on the metabolism of plants...

It is therefore impossible to endorse the publication in the journal.

Author Response

Thanks for your comments

ü  Lack of analysis on the Si content and other soil macro and microelements,

- In current situation, we can't analyze it.

ü  Lack of assessment of the different climatic conditions of the two locations on the determined parameters,

- I guess we fully provide climatic conditions of both locations in current manuscript.

ü  Impossible methods of determining the roots growth by in situ experiments and partially harvesting them,

- In the crown root analysis, root washing methods are very common technology. Using same methods, we published many paper in MDPI as well as reputative SCI(E) journal. 

ü  The determination of the quantity of chlorophyll with the chlorophyll meter is an estimative method,
- Many researchers are used chlorophyll contents by chlorophyll meter. In particular, CCM-100 is more advanced chlorophyll meter to detect chlorophyll contents in various crops.

ü  Morphological determinations used by measurement and weighing... ???

- Sorry, I couldn't understand your comment.

ü  There are not used modern experimental techniques, biochemical analysis to be able to quantify the impact of the application on the metabolism of plants...

- I guess you didn't fully understood purpose of this manuscript. If purpose of this manuscript was an identification of Si fertilizer on the metabolism of soybean we submit this manuscript to other journal.